# Unsupervised Class-Incremental Learning through Confusion

## Abstract

While many works on Continual Learning have shown promising results for mitigating catastrophic forgetting, they have relied on supervised training. To successfully learn in a label-agnostic incremental setting, a model must distinguish between learned and novel classes to properly include samples for training. We introduce a novelty detection method that leverages network confusion caused by training incoming data as a new class. We found that incorporating a class-imbalance during this detection method substantially enhances performance. The effectiveness of our approach is demonstrated across a set of image classification benchmarks: MNIST, SVHN, CIFAR-10, CIFAR-100, and CRIB.

## 1 Introduction

The development of continually learning systems remains to be a major obstacle in the field of artificial intelligence. The primary challenge is to mitigate catastrophic forgetting: learning new tasks while maintaining the ability to perform old ones. This domain of research is often referred to as Continual Learning, Lifelong Learning, Sequential Learning, or Incremental Learning: each with subtleties in the learning environment and training process, but most with the use of supervision (De Lange et al. (2020)).

Recently, Stojanov et al. (2019) introduced a novel unsupervised class-incremental learning problem motivated by the desire to simulate how children's play behaviors support their ability to learn object models (IOLfCV). Here, sequential tasks take the form of exposures. Each exposure is comprised of a set of images that pertains to a single class that is hidden from the learner. Exposure *boundaries*, the transition from one exposure to the next, are known. The model trained in this setting, is analogous to a young child that has been placed in a playpen with a set of new toys. The child steadily gathers information over time by picking up, examining, and putting down new/old objects continuously. Similar to how the child does not have any guidance to the toys they will examine, the agent does not have access to the exposure's identity during training.

To learn in the unsupervised class-incremental setting, an agent must conduct two procedures successfully. Given a new learning exposure, the key step is to perform novelty detection: to identify whether an exposure corresponds to a class that has been learned. If the agent determines that an exposure is familiar, the second step is to identify its label such that the exposure can be leveraged to update the model. Both procedures must be performed reliably. Otherwise, the novelty detection mistakes will result in label noise that distorts the learned model, increasing the likelihood of subsequent mistakes.

Deep neural networks are known to make overconfident decisions for anomalous data distributions that were not seen during training (Hendrycks & Gimpel (2016)). To address this problem, research related to out-of-distribution (OOD) detection have utilized supervised methods (Liang et al. (2017); Alemi et al. (2018)) and unsupervised methods (Choi & Jang (2018); Hendrycks et al. (2018); Serrà et al. (2019)). Works related to open set recognition have also addressed the OOD problem by applying distance-based thresholds computed from known class scores (Scheirer et al. (2012; 2014)). The work by Stojanov et al. (2019) applies a similar method to the unsupervised incremental setting by computing class features produced from a set of supervised samples. In contrast, we propose a model, Incremental Learning by Accuracy Performance (iLAP), that determines class novelty and identity by considering performance changes of previously learned tasks when an incoming set of exposure images are trained under a new label.

Instead of using a distance-based metric, our novelty detection threshold relies on the percentage of accuracy that was maintained by performing a model update using the incoming exposure. This poses several practical advantages: First, the threshold value does not rely on supervised samples and is more intuitive (Section 3.3). Second, the performance of our method is independent of the sequence of the incoming exposure classes (Section 5.2). Finally, the model is able to distinguish between *similar* classes more reliably (Section 5.3).

From our experiments, we demonstrate that the confusion resulting from training with label ambiguity results in a more reliable signal for novelty detection in comparison to previous methods. We demonstrate that our technique is more robust and results in substantial performance gains in comparison to various baselines. Furthermore, despite the absence of labels, our model was able to perform similarly to supervised models under several benchmarks.

In summary, this work provides three contributions:

- We present a novel framework, iLAP, that achieves learning in the unsupervised class-incremental environment where the exposure identities are unknown.

- We demonstrate that by including a class-imbalance technique, our unsupervised method is able to closely match supervised performance for several image classification benchmarks trained in the incremental setting.

- We identify failure cases that are overlooked by traditional OOD methods that leverage distance-based thresholds.

## 2 RELATED WORKS

Introduced by Stojanov et al. (2019), the unsupervised class-incremental setting contains a set of sequential tasks that are single-class exposures; classes pertaining to the exposures may repeat and are unknown. This is not to be mistaken with unsupervised continual learning (UCL) where task boundaries and task identities are unavailable (Lee et al. (2020); Smith & Dovrolis (2019); Rao et al. (2019)). Our work presents an agent that is able to leverage the *boundary* information from the unsupervised class-incremental environment to achieve performances that are close to models trained under supervision.

### 2.1 CONTINUAL LEARNING/INCREMENTAL LEARNING

Prior works in this field primarily aim to improve a model's ability to retain information while incorporating new tasks (Goodfellow et al. (2013); Parisi et al. (2019); Rebuffi et al. (2017); Lopez-Paz & Ranzato (2017); Aljundi et al. (2018); Castro et al. (2018)). Typically, these models reside in learning settings where both task labels and task boundaries are available. Methods include replay techniques, the usage of artifacts and generated samples to refresh a model's memory (Kamra et al. (2017); Wu et al. (2018); Rolnick et al. (2019); Shin et al. (2017); Wu et al. (2019)), and regularization-based practices, the identification and preservation of weights that are crucial for the performance of specific tasks (Kirkpatrick et al. (2017); Zenke et al. (2017); Yoon et al. (2017)). In contrast to prior works, our method addresses incremental learning in a setting where exposure labels are unavailable.

### 2.2 UNSUPERVISED CONTINUAL LEARNING

Recently, a series of works tackle the UCL problem where task boundaries and task identities are unknown. Smith & Dovrolis (2019) performs novelty detection by analyzing an input image through a series of receptive fields to determine if an input patch is an outlier. Meanwhile, CURL proposes a method to learn class-discriminative representations through a set of shared parameters (Rao et al. (2019)). CN-DPM, introduces an expansion-based approach that utilizes a mixture of experts to learn feature representations (Lee et al. (2020)). Although CN-DPM performs in a *task-free* setting, incoming tasks are multi-class and individual class labels are provided. This supervision is required to train the existing experts and determine when a new one is needed. While boundary information is not required for these works, the performances are far below supervised baselines (77.7% on and MNIST 13.2% Omniglot) (Rao et al. (2019)).

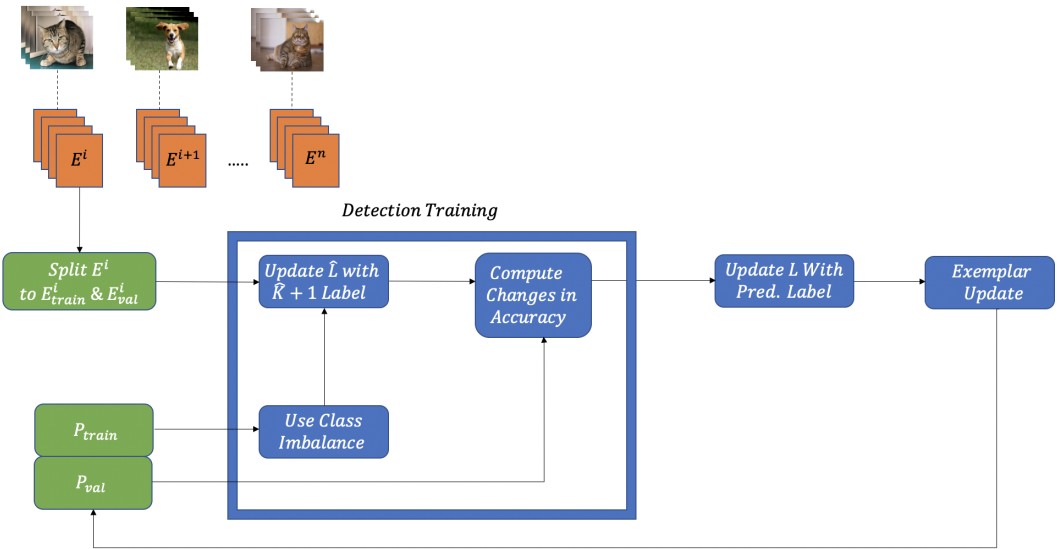

Figure 1: Incoming exposures are split into $E_{train}^i$ and $E_{val}^i$. $E_{train}^i$ is aggregated with $\mathcal{P}_{train}$ and is used for detection training. $\hat{L}$ is updated with samples from $E_{train}^i$ labeled as $\hat{K} + 1$. The change in model accuracy is assessed using $\mathcal{P}_{val}$ to identify novelty and class identity. Finally, $L$ is updated with the predicted label; samples from $E_{train}^i$ and $E_{val}^i$ are saved to their respective exemplars.

## 2.3 OUT-OF-DISTRIBUTION DETECTION

This ongoing area of research aims to detect outliers in training and testing data. Current approaches can be largely categorized by statistical, distance-based, and deep learning methods (Eskin (2000); Yamanishi et al. (2004); Knorr et al. (2000); Hautamaki et al. (2004); Sabokrou et al. (2018); Kliger & Fleishman (2018)). Recent techniques involve using a threshold to determine class novelty from network confidence values (Hendrycks & Gimpel (2016)). ODIN uses input perturbations to increase softmax scores for neural networks to distinguish in-distribution images from out-of-distribution images (Liang et al. (2017)). DeVries & Taylor (2018) incorporates a confidence branch to obtain out-of-distribution estimations. Our method (iLAP) is the first to incorporate a threshold value that is dependent on class-accuracy changes caused by data poisoning.

## 3 APPROACH

In this section, we provide an overview of our method. We begin by identifying the learning setting, followed by details of our training process. Finally, insights for choosing threshold values are provided.

### 3.1 SETTING

In the unsupervised class-incremental setting, a learner $L$ perceives an input stream of exposures denoted as $E^1, E^2, ..., E^n$. Each exposure contains a set of images, $E^i = \{e_1^i, e_2^i, ..., e_{n_i}^i\}$, $e_j^i \in \mathbb{R}^{C \times H \times W}$, where $C$, $H$, and $W$ are the number of channels, height, and width of the input image respectively. Each exposure belongs to a single class $y_i \in \mathbb{N}$, which has been sampled from class distribution $P(\mathcal{C})$. For each $E^i$, $L$ does not have knowledge of the true class label $y_i$. Two exemplars, $\mathcal{P}_{train} = (P_{train}^1, P_{train}^2, ..., P_{train}^{\hat{K}})$ and $\mathcal{P}_{val} = (P_{val}^1, P_{val}^2, ..., P_{val}^{\hat{K}})$, are maintained at all times, where $\hat{K}$ denotes the total number of classes $L$ has currently determined. The exemplars are used to store samples from the exposure for replay and accuracy assessment. The sizes of both exemplars, $\left|P_{train}^i\right|$ and $\left|P_{val}^i\right|$, are bounded per class.

### 3.2 DETECTION TRAINING

For each incoming exposure, the model is tasked to determine whether the class associated with the exposure was learned previously. Our solution is to perform a model update by treating the incoming exposure as a new class, we coin this technique *detection training*. Under the circumstances that the exposure class is repeated, the performance for the previously learned class would suffer drastically after training. The reason for this behavior is because the model has associated two different labels to a similar class distribution.

During detection training, $\hat{L}$, a copy of $L$ is produced. The incoming exposure is assigned with label $\hat{K} + 1$. Train-validation split is performed on the incoming exposure to obtain $E_{train}$ and $E_{val}$, and are aggregated with exemplars $\mathcal{P}_{train}$ and $\mathcal{P}_{val}$ respectively. The combined samples are used to train $\hat{L}$ via validation-based early stopping. We denote the vector $\{\Delta_{\hat{y}}\}_{\hat{y} \in [\hat{K}]}$ to represent the percentage decrease of the class accuracies (computed using $\mathcal{P}_{val}$) before and after the update. If $\max(\{\Delta_{\hat{y}}\})$ exceeds a threshold, $\theta$, the incoming exposure is likely to have been learned by $L$. In this case, the correct identity pertaining to the exposure is $\arg\max_{\hat{y} \in [\hat{K}]} \Delta_{\hat{y}}$. Otherwise, if $\theta$ is unsatisfied, $\hat{K} + 1$ is the appropriate label for the new class.

### 3.3 CLASS-IMBALANCE FOR DETECTION TRAINING

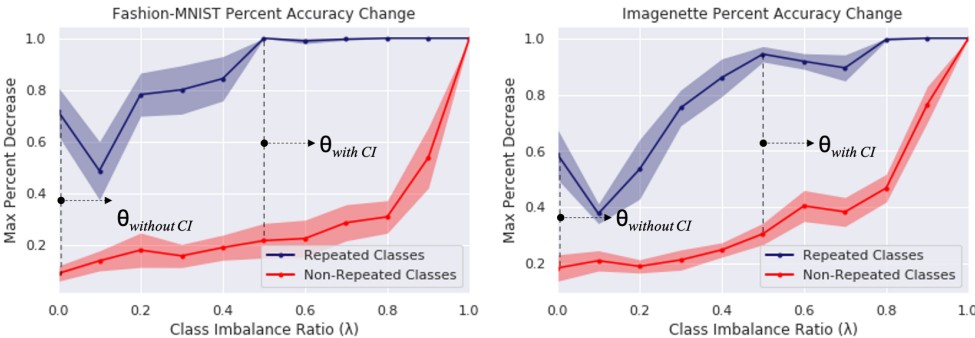

Figure 2: Graphs depicting the average accuracy decrease for repeated vs. non-repeated classes at various values of class-imbalance. As the ratio of class-imbalance increases, the accuracy change for non-repeated classes remains low, with the exception at high ratios when very few samples are present during the model update (red). The accuracy decrease for repeated classes is amplified as the ratio of class-imbalance increases (blue).

Introducing a class-imbalance during detection training creates a more distinct decision boundary by exacerbating the class-accuracy drop for repeated exposures. Consider a theoretical case where an optimal model has learned $\hat{K}$ classes. The incoming exposure, $E^i$, contains a distribution that is equal to that of some previously learned class $\hat{y}_i$. If the model were updated with equal samples of $E^i$ labeled $\hat{K} + 1$, and $P_{train}^{\hat{y}_i}$ labeled $\hat{y}_i$, the accuracy for class $\hat{y}_i$ would become ambiguous during validation ($\hat{y}_i \approx 50\%$, $\hat{K} + 1 \approx 50\%$). However, if the model were updated with a greater sample of $\hat{K} + 1$ labels, the accuracy drop for class $\hat{y}_i$ would be considerably larger because $\hat{K} + 1$ would be favored during inference.

$\hat{L}$ has the option to use an imbalanced dataset during detection training where a fraction of the size for each class from $\mathcal{P}_{train}$ are used in comparison to the size of $E_{train}$. Let $P_{sampled}^i \subset P_{train}^i$, the class-imbalance ratio $\lambda$ is:

$$\lambda = 1 - \frac{|P_{sampled}^i|}{|E_{train}|} \tag{1}$$

In **Figure 2**, we use Fashion-MNIST (Xiao et al. (2017)) and Imagenette (Howard) to obtain a general value for $\lambda$ and $\theta$ for our experiments with other datasets. Performance drops are tracked for the repeated class and non-repeated classes during detection training with various values of $\lambda$. The

goal is to maximize the performance loss for the repeated class while maintaining the performance for the other classes. In **Section 4** a $\theta$ value of $0.6$ and a $\lambda$ value of $0.5$ are used. $\lambda$ is determined by the point of the maximum distance between the two accuracy curves; $\theta$, the expected accuracy drop for a repeated exposure, is the mean of the two values at $\lambda$ **Figure 2**.

### 3.4 MODEL AND EXEMPLAR UPDATE

After obtaining the predicted label ($\hat{K} + 1$ or $\arg\max_{\hat{y} \in [\hat{K}]} \Delta_{\hat{y}}$) from detection training, $L$ is trained with the aggregated dataset obtained from $P_{train}$ and $E_{train}$. Two sets with the most representative samples from $E_{train}$ and $E_{test}$ are created and added to their respective exemplars for future replay. The selection process is determined by ranking the distance between the image features and the class-mean image features. This methodology is consistent with the procedure introduced by Castro et al. (2018).

The availability of $\mathcal{P}_{val}$ allows $L$ to assess how well old and new classes are considered. If the accuracy for any class interpreted by $L$ falls below a percentage, the class is altogether discarded. This allows the model to remove extraneous classes that were learned insufficiently or have been forgotten. In **Section 4**, a percentage threshold of 20% is used for all experiments.

## 4 EXPERIMENTS

The performance of our framework is evaluated using a series of image classification benchmarks: MNIST, SVHN, CIFAR-10, CIFAR-100, and CRIB (LeCun (1998); Netzer et al. (2011); Krizhevsky et al. (2009); Stojanov et al. (2019)). First, we compare our novelty detector to related OOD methods. Next, we evaluate performance of iLAP to that of other incremental learners: BiC (Wu et al. (2019)) (supervised) and IOLfCV (unsupervised).

### 4.1 EXPERIMENTAL DETAILS

For the following experiments, a ResNet-18 model (He et al. (2016)) pre-trained with ImageNet is used for iLAP and all baselines (additional experiments without pretraining are presented in **Appendix A.4**). $\lambda = 0.5$ and $\theta = 0.6$ are used for iLAP with class-imbalance detection training, while a $\lambda = 0$ and $\theta = 0.4$ are used for iLAP without class-imbalance detection training. The parameters are maintained across all benchmarks.

For each exposure, the model is trained for 15 epochs with 16 batch size, using an Adam (Kingma & Ba (2014)) optimizer with validation-based early stopping; a learning rate of $2e-4$ is used. The feature extraction layers use a ten times lower learning rate at $2e-5$. For all models, the exemplar size per class is equal to the exposure size. The exposure validation split ratio is 0.8 (e.g. for exposure size = 200, iLAP: $[E_{train}] = 160$ and $[E_{val}] = 40$). The thresholds used for IOLfCV in **4.1** were determined by maximizing the F-score for the classification of in-distribution exposures versus out-of-distribution exposures. To obtain the best performance possible, the entirety of the dataset was used. The values are $0.46$, $0.63$, $0.57$, and $0.62$ for MNIST, SVHN, CIFAR-10, and CIFAR-100 respectively.

### 4.2 OUT-OF-DISTRIBUTION DETECTION RESULTS

The OOD detectors are assessed in an incremental setting with size 200 exposures. The detectors are evaluated on their ability to determine whether an exposure is novel by using the common established metrics: FPR95, AUROC, and AUPR (Hendrycks & Gimpel (2016)). Details of the compared works and evaluation methods are described in **Appendix A.1**. The results are illustrated in **Table 1**.

| | MNIST | | | CIFAR-10 | | |
|---|---|---|---|---|---|---|
| | FPR95↓ | AUROC↑ | AUPR↑ | FPR95↓ | AUROC↑ | AUPR↑ |
| MSP | 0.14±.09 | 0.98±.01 | 0.95±.02 | 0.60±.13 | 0.86±.04 | 0.71±.09 |
| CE | 0.35±.13 | 0.80±.04 | 0.67±.09 | 0.64±.13 | 0.78±.04 | 0.54±.03 |
| ODIN | 0.12±.09 | 0.98±.01 | 0.95±.02 | 0.55±.12 | 0.87±.04 | 0.71±.09 |
| ZeroShot OOD | 0.03±.03 | 0.99±.01 | 0.99±.01 | 0.63±.12 | 0.67±.06 | 0.47±.08 |
| IOLfCV | 0.12±.08 | 0.98±.01 | 0.95±.02 | 0.60±.12 | 0.78±.06 | 0.60±.08 |
| iLAP w/o CI (Ours) | 0.19±.02 | 0.92±.02 | 0.68±.05 | 0.58±.03 | 0.69±.03 | 0.40±.05 |
| iLAP w/ CI (Ours) | **0.0 ± .0** | **1.0 ± .0** | **1.0 ± .0** | **0.32 ± .10** | **0.93 ± .03** | **0.81 ± .08** |

| | SVHN | | | CIFAR-100 | | |
|---|---|---|---|---|---|---|
| | FPR95↓ | AUROC↑ | AUPR↑ | FPR95↓ | AUROC↑ | AUPR↑ |
| MSP | 0.08±.04 | 0.98±.01 | 0.98±.01 | 0.20±.04 | 0.95±.02 | 0.71±.09 |
| CE | 0.5 ± .1 | 0.78±.03 | 0.72±.07 | 0.55±.12 | 0.82±.01 | 0.63±.02 |
| ODIN | 0.12±.01 | 0.94±.01 | 0.93±.02 | 0.21±.04 | 0.94±.02 | 0.91±.01 |
| ZeroShot OOD | 0.06±.04 | 0.97±.01 | 0.98±.01 | 0.23±.03 | 0.97±.01 | 0.94±.02 |
| IOLfCV | 0.28±.11 | 0.96±.01 | 0.95 ± .0 | 0.28±.05 | 0.96±.01 | 0.94±.01 |
| iLAP w/o CI (Ours) | 0.08±.02 | 0.92±.02 | 0.89±.04 | 0.27±.05 | 0.92±.05 | 0.72±.12 |
| iLAP w/ CI (Ours) | **0.0 ± .0** | **0.99 ± .0** | **0.98 ± .0** | **.08 ± .02** | **0.99 ± .0** | **0.98 ± .0** |

Table 1: Comparison of our technique, with and without class-imbalance, to recent novelty detection methods. FPR95, AUROC, and AUPR are used to evaluate performance. Our method is the most effective in the incremental learning setting where exposure samples are limited.

| Test Accuracy (%) | | | | | |
|---|---|---|---|---|---|
| | MNIST | SVHN | CIFAR-10 | CIFAR-100 | CRIB |
| BiC (Supervised) | **98.0 ± 0.2** | **89.2 ± 0.3** | **74.9 ± 0.4** | **72.5 ± 0.3** | **88.5 ± 0.2** |
| IOLfCV (Unsupervised) | 88.0 ± 3.5 | 46.8 ± 8.6 | 45.3 ± 4.7 | 61.6 ± 0.6 | 66.8 ± 2.1 |
| iLAP w/o CI (Ours) | 90.3 ± 1.7 | 84.4 ± 1.7 | 60.9 ± 1.4 | 65.0 ± 1.2 | 59.0 ± 1.5 |
| iLAP w/ CI (Ours) | **98.0 ± 0.2** | **88.1 ± 0.8** | **67.4 ± 1.7** | **68.6 ± 0.7** | **67.8 ± 1.7** |

Table 2: Comparison of iLAP to IOLfCV and BiC by test accuracy (%) using the MNIST, SVHN, CIFAR-10, CIFAR-100 and CRIB datasets.

| No. of Unique Classes Learned | | | | | |
|---|---|---|---|---|---|
| | MNIST | SVHN | CIFAR-10 | CIFAR-100 | CRIB |
| BiC (Supervised) | **10.0 ± 0.0** | **10.0 ± 0.0** | **10.0 ± 0.0** | **100.0 ± 0.0** | **50.0 ± 0.0** |
| IOLfCV (Unsupervised) | 9.2 ± 0.4 | 8.8 ± 1.2 | 8.7 ± 0.7 | 89.0 ± 0.5 | 42.0 ± 0.3 |
| iLAP w/o CI (Ours) | 10.0 ± 0.0 | 10.0 ± 0.0 | 10.0 ± 0.0 | 96.0 ± 0.5 | 43.0 ± 0.1 |
| iLAP w/ CI (Ours) | **10.0 ± 0.0** | **10.0 ± 0.0** | **10.0 ± 0.0** | **96.5 ± 0.5** | **45.0 ± 0.2** |

Table 3: Comparison of iLAP to IOLfCV and BiC by the number of unique classes learned using the MNIST, SVHN, CIFAR-10, CIFAR-100 and CRIB datasets.

### 4.3 INCREMENTAL LEARNING RESULTS

The accuracy of learner $L$ is computed using the ground-truth mapping $m : [\hat{K}] \to [K]$ with the equations:

$$S(x, y) = \begin{cases} \frac{1}{|m^{-1}(y)|}, & \text{if } L(x) \in m^{-1}(y) \\ 0, & \text{otherwise} \end{cases}$$

$$Accuracy = \mathbb{E}_{x,y \sim test} [S(x, y)]$$

The learner accuracy is the mean of the sample accuracy scores evaluated on the test set, where $(x, y)$ represents a single sample. For each sample with label $y$, let $m^{-1}(y)$ represent the corresponding labels from the learner. In the case that a learner output does not belong to set $m^{-1}(y)$, an accuracy score of $0$ is assigned (class is not detected). Otherwise, an accuracy score of $\frac{1}{|m^{-1}(y)|}$ is designated to penalize the learner if $m$ is non-injective and have attributed multiple labels to a single ground truth class. Performance results are illustrated in **Table 2 & 3**. Additional visualizations are provided in **Appendix A.2, A.3** and **A.4**.

## 5 ANALYSIS

Traditional OOD methods that rely on distance-based thresholds are restricted by the supervised samples that are available. These values are non-intuitive and vary drastically across datasets (whereas our percentage threshold are $\approx 50\%$ for all datasets). In the incremental learning setting, early mistakes are amplified as more exposures are introduced, a proper threshold initialization dictates a model's feasibility. However, we argue that even with a *good* threshold these methods will consistently fail in particular conditions. The purpose of this section is to discuss the results obtained from our experiments. Subsequently, we highlight a few cases that are overlooked by traditional distance-based methods.

### 5.1 OUT-OF-DISTRIBUTION DETECTION ANALYSIS

iLAP with class-imbalance detection training (CI) was able to outperform related OOD methods for the MNIST, SVHN, CIFAR-10, and CIFAR-100 benchmarks under all metrics **Table 1**. However, the results for iLAP without CI were not as definitive. CE performed the worst in the incremental setting, possibly because the performance of the confidence branch is reliant on larger training samples. IOLfCV's method performed on par with related methods.

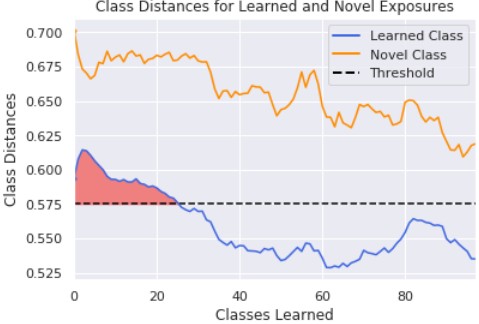

Figure 3: Visualization depicting the behavior of class feature distances as more classes are learned by the network. The optimal threshold computed by optimizing the F-score is the midpoint of the learned and novel class values when all 100 classes are incorporated. However, this threshold will mistakenly recognize repeated classes as new classes early on during incremental training.

## 5.2 UNSUPERVISED INCREMENTATL LEARNING ANALYSIS

iLAP with CI was able to beat IOLfCV by 10.0, 41.3, 22.1, 7.0, and 1.0 percentage points for the MNIST, SVHN, CIFAR-10, CIFAR-100, and CRIB benchmarks respectively **Table 2**. iLAP was also able to maintain its performance when exposure sizes are decreased **Appendix A.3**. We found that the reason for the lower performance beat for CIFAR-100 and CRIB is not directly attributed to the larger number of classes in the dataset. Rather, the problem lies with how the exposure sequence for the incremental learning setting is created and how the distance-based threshold is calculated.

The threshold for IOLfCV is computed by maximizing the F-score for the binary classification of novel versus non-novel classes. In our experiments, the entirety of the dataset was used to compute the baseline's threshold. Although this is impractical, we wanted to illustrate that iLAP is able to beat the baseline even under the most optimal conditions.

In **Figure 3** we illustrate the behavior of the class feature distances as more classes are incorporated by a network. The most optimal threshold that maximizes the F-score lies in the mid-point between the two graphs when all 100 classes are learned. However, because the threshold is fixed, the novelty detector fails to correctly identify repeated classes early on during training and is more inclined to label repeated classes as unseen, (shaded red area in **Figure 3**). Consistent with the described setting in Stojanov et al. (2019), each class within a benchmark is repeated an equal number of times in a randomized sequence. For datasets with a large number of classes, there is a higher chance that the repeated exposures are further apart. Therefore, IOLfCV seemingly performs comparatively better on CIFAR-100 than CIFAR-10, but would fail if early repeated exposures were frequent.

## 5.3 CLASS SIMILARITY

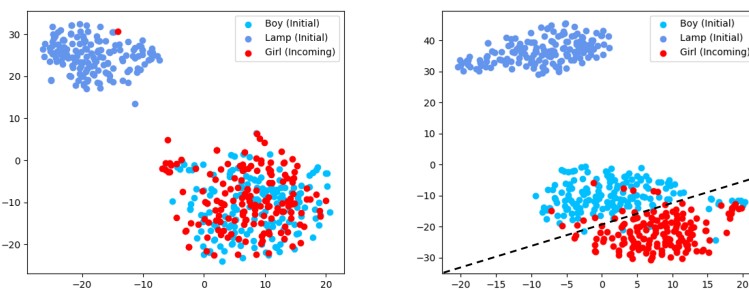

Figure 4: TSNE plot depicting a model which has been trained to classify *lamp* vs. *boy*. An incoming exposure with class *girl* is introduced to the model. Comparing the feature distributions of this model, *boy* and *girl* have almost identical distributions; the OOD detector mistakenly assesses them to be same class (left). In contrast, our method first attempts to learn a feature space where the two classes are separable and identifies *boy* and *girl* correctly (right).

iLAP was able to detect all classes for MNIST, SVHN, and CIFAR-10 and on average 96.5 out of 100 classes for CIFAR-100. Meanwhile, IOLfCV struggles to identify unique classes for all evaluated benchmarks. Through closer inspection, we found that the distance-based method is unable to distinguish classes when they are too *similar*.

Consider two classes, $k_1$ and $k_2$, that can be separated by a classifier $\mathcal{C}$ in learned feature space $\mathcal{F}$. An incoming exposure, class $k_3$, shares a similar distribution to class $k_1$ in feature space $\mathcal{F}$, but separable in some feature space $\mathcal{F}'$. The distance-based method is highly probable to fail because it is likely to categorize $k_1$ and $k_3$ as identical classes. However, because our method always trains the incoming exposure as a new class, $\mathcal{C}$ is forced to learn the feature space $\mathcal{F}'$ in which these two classes are separable. **Figure 4** illustrates two prior classes, *boy* and *lamp*, in some feature space. The incoming exposure, class *girl*, is unable to be distinguished from class *boy* by the distance-based method **Figure 4 (left)**. However, because detection training always attempts to classify the incoming exposure as a new class, our method is able to identify $\mathcal{F}'$ **Figure 4 (right)**.

## 6 CONCLUSION

To achieve learning in an unsupervised class-incremental setting, a reliable novelty detector is needed. Current methods utilize a detection threshold that is calibrated using class feature distances. In our work, we illustrate that the use of a static distance-based threshold is not only impractical but also unreliable. Instead, we introduce a technique that leverages confusion error to perform novelty detection by always training the incoming exposure as a new class. Using a series of image classification benchmarks, we illustrate that our method is able to closely rival supervised performance despite the lack of labels.

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

# A    APPENDIX

## A.1    OOD EXPERIMENTAL DETAILS

A brief description of the evaluation metrics used for **Table 1**.

- **False Positive Rate at 95% (FPR95)**: This measure determines the False Positive Rate (FPR) when True Positive Rate (TPR) is equal to $95\%$. FPR is calculated as $\frac{FP}{FP+TN}$ where FP is the number of False Positives and TN is the number of True Negatives. TPR is calculated as $\frac{TP}{TP+FN}$ where TP is the number of True Positives and FN is the number of False Negatives.

- **Area Under the Receiver Operating Characteristic (AUROC)**: This metric illustrates the relationship between the TPR and the FPR. This measure determines the probability that a novel class will have a higher detection score compared to a non-novel class.

- **Area Under the Precision-Recall (AUPR)**: This metric is constructed by plotting precision versus recall. The AUPR curve treats the novel examples as the positive class. A high area value represents high recall and high precision.

The training detection method used in iLAP is compared to a set of related works in OOD. The following works reflect the acronyms used in **Table 1**

- **MSP (Hendrycks & Gimpel (2016))**: MSP is computed from a trained classifier to perform out-of-distribution detection. The mean MSP for all images belonging to an exposure is used to determine novelty.

- **CE (DeVries & Taylor (2018))**: CE requires extending a model with a confidence branch to obtain a set of values. These values reflect the model's ability to produce a correct prediction. When the mean estimation value for a set of input images is low, the sample is likely to be novel.

- **ODIN (Liang et al. (2017))**: ODIN uses temperature scaling and input perturbations to widen the MSP difference between in-distribution and out-of-distribution samples. The optimal value for temperature and perturbation magnitude are found by minimizing FPR using grid search. 2 and 0.0012 are used for temperature and perturbation magnitude values respectively, for both the MNIST and CIFAR-10 dataset.

- **Zero-Shot OOD (Sastry & Oore (2019))**: Zero-Shot OOD uses pairwise correlations of network features to detect novel samples. The class-conditional feature correlations, on

the training data, are computed across all layers of the network. The mean correlations are then compared with the pairwise mean feature correlations from a test sample to obtain a deviation value.

- **IOLfCV (Stojanov et al. (2019))**: IOLfCV determines a distance-based threshold computed using average-feature means from a network trained from supervised samples. Two initialized classes are used to compute the threshold by finding the optimal point using precision-recall analysis.

## A.2 MAIN EXPERIMENT RESULTS

The following are produced with the use of a GTX TITAN X gpu. For each exposure, the model is trained for 15 epochs with 16 batch size, using an Adam optimizer. The learning rate used is $2\mathrm{e}{-4}$. The feature extraction layers use a ten times lower learning rate at $2\mathrm{e}{-5}$. The input size is 224.

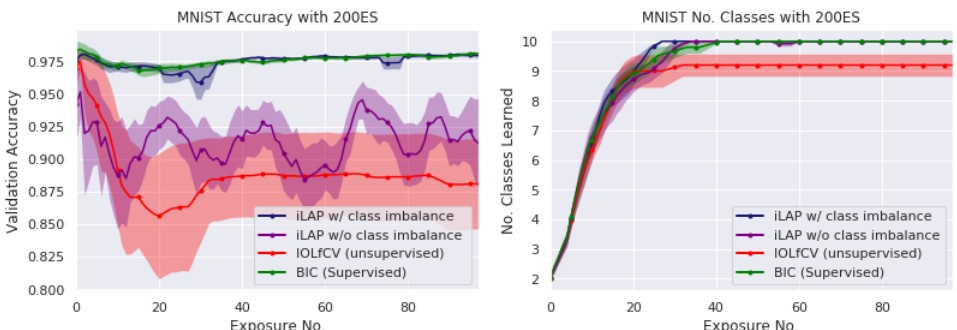

Figure 5: Visualizations comparing accuracy (left) and number of classes detected (right) for the MNIST benchmark.

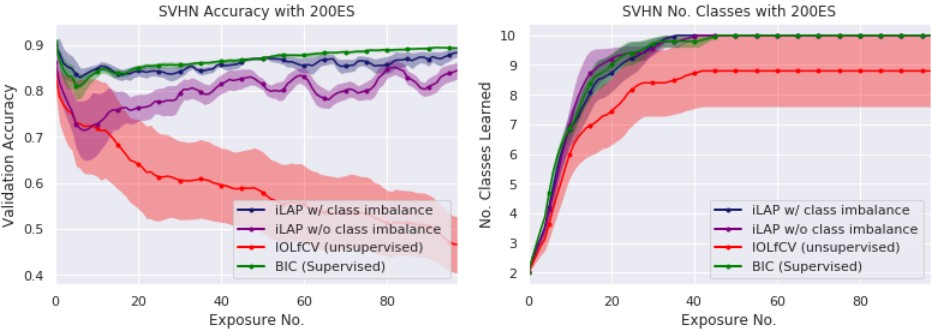

Figure 6: Visualizations comparing accuracy (left) and number of classes detected (right) for the SVHN benchmark.

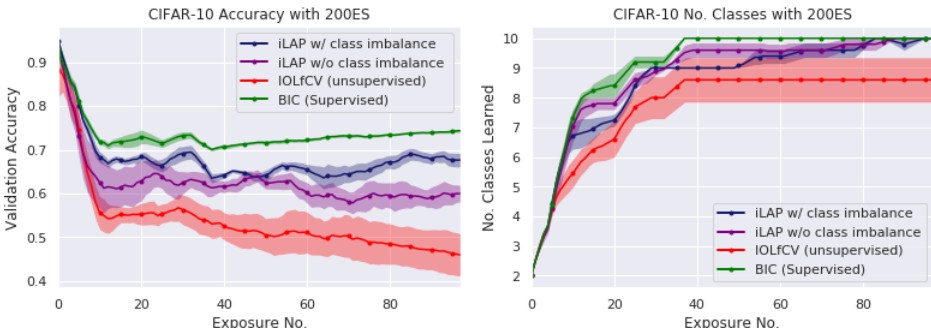

Figure 7: Visualizations comparing accuracy (left) and number of classes detected (right) for the CIFAR-10 benchmark.

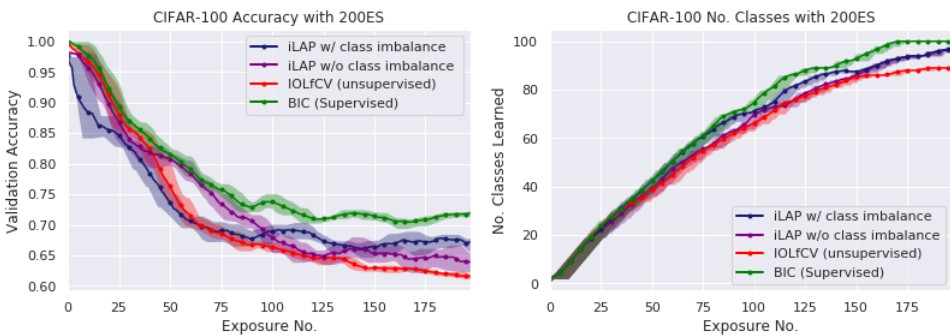

Figure 8: Visualizations comparing accuracy (left) and number of classes detected (right) for the CIFAR-100 benchmark.

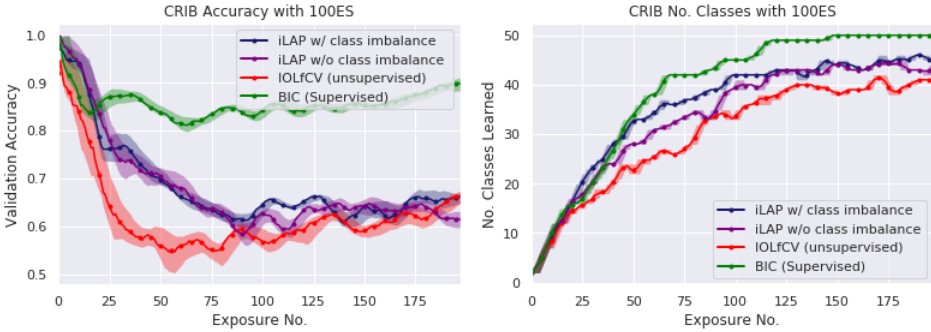

Figure 9: Visualizations comparing accuracy (left) and number of classes detected (right) for the CRIB benchmark.

## A.3 EXPERIMENTS WITH LOWER EXEMPLAR SIZES

In this section, we showcase iLAP's results at lower exposure sizes.

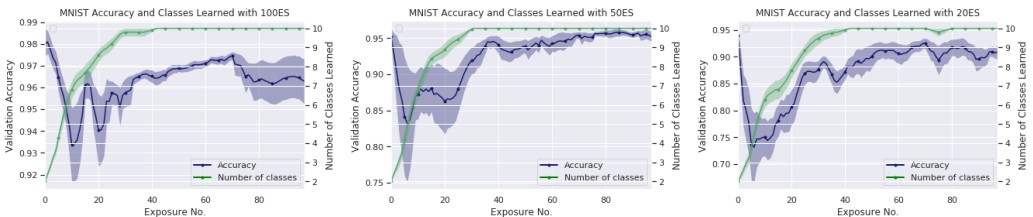

Figure 10: Visualizations depicting iLAP's performance on the MNIST benchmark with 100, 50, and 20 exemplar sizes (left to right).

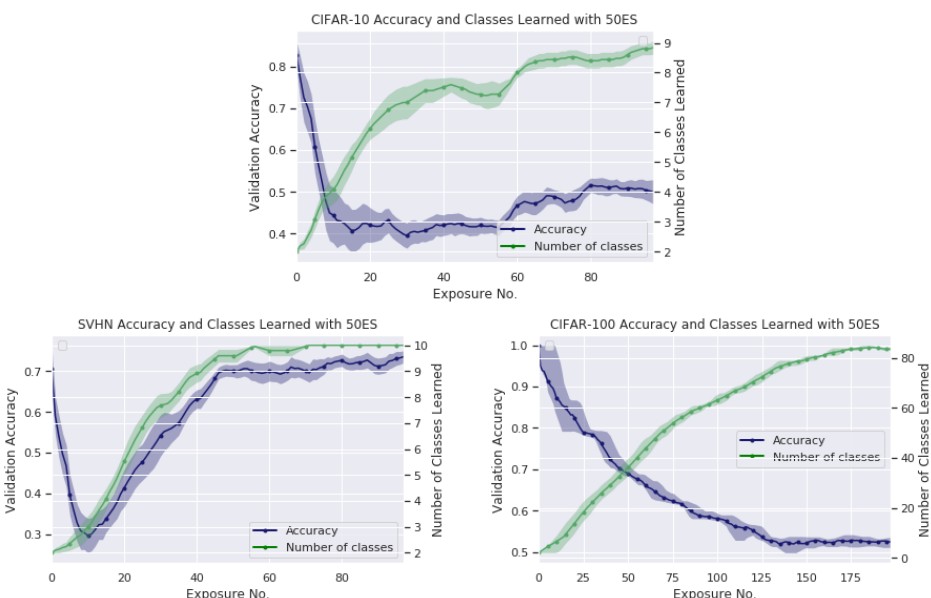

Figure 11: Visualizations depicting iLAP's performance on the SVHN, CIFAR-10, and CIFAR-100 benchmark with a exemplar size of 50.

## A.4   EXPERIMENTS WITHOUT PRE-TRAINING

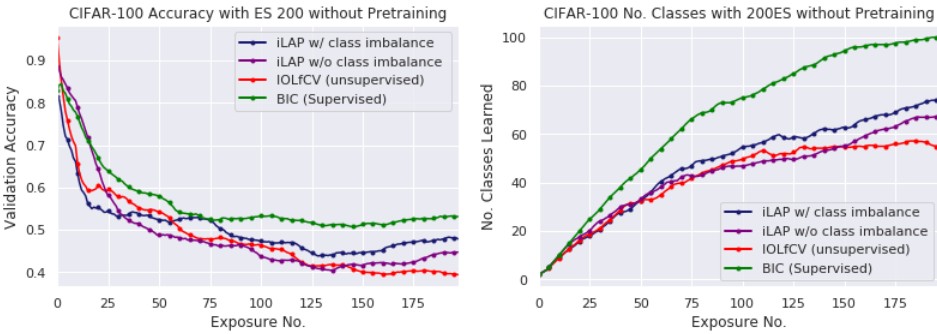

Figure 12: Visualizations comparing accuracy (left) and number of classes detected (right) for the CIFAR-100 benchmark without the use of pre-trained weights.

