# OpenReview forum: "Unsupervised Class-Incremental Learning through Confusion"
_ICLR.cc/2021/Conference — Reject_

### Official Review · AnonReviewer2 · 2020-10-19
**Lacking proper motivation and missing fair comparison**

**Rating:** 3
**Confidence:** 3

**Review:**

The authors propose a novelty detection module to help unsupervised class-incremental learning. The novelty detection relies on the percentage of accuracy drop during a model update when treating incoming data as a new class. If the model maintains high accuracy, then the module treats the incoming data as familiar, thereby choosing one of the existing classes as the correct label. The paper investigates the effectiveness of the proposed method on MNIST, SVHN, CIFAR-10, and CIFAR-100.

The main weakness of the submission might be that the proposed novelty detection is not well motivated. The bottleneck of the proposed pipeline comes down to whether the accuracy drop on the selected subset is a good indicator of out-of-distribution (OOD) detection. The submission does not provide theoretical insights nor direct references that show it is actually the case. In fact, in the experimental section (Sec 4.1), the authors have to use several different accuracy threshold settings (0.46, 0.63, 0.57, 0.62) for different datasets, demonstrating that accuracy drop might not be a reliable indicator.

Besides, the direct competing method CURL (Rao et al.) is cited but not compared. iCarl [R1] should be quite related as well. The authors also use quite a different backbone network (ResNet-18) than other competing methods. Therefore, it is hard to justify whether the proposed approach is more effective than other baselines.

The method described here is also quite similar to the field of active learning. It would be great to discuss the relationship between the proposed novelty detection and other active learning literatures.

Sec 1 first sentence “continually learning systems remains to be a major obstacle in the field of artificial intelligence” is quite a strong statement. I believe there are other major obstacles in AI and they should be discussed as well.

Sec 1 paragraph 3, “an agent must conduct two procedures successfully”. The authors do not clearly define what is an “agent” in the context. It is hard for the readers to follow through the manuscript.

Sec 3.4, “After obtaining the correct label”. Actually the label technically is not “correct” but assumed to be correct for the class-incremental learning.

[R1] Rebuffi et al. Icarl: Incremental classifier and representation learning. In CVPR 2017.

---

> ### Author Response · Authors · 2020-11-11
> **Re: Lacking proper motivation and missing fair comparison**
>
> Thank you for your comments. In section 4.1, the threshold settings (0.46, 0.63, 0.57, 0.62) refer to the baseline method IOLfCV, not our method. These values were determined by maximizing the F-score for the binary classification task for novel vs non-novel classes across the entirety of each dataset. Meanwhile, we use a fixed accuracy threshold of ($\theta
> $) 0.6  with class imbalance ratio of ($\lambda$) 0.5  for all benchmarks presented (paragraph 1, section 4.1). It is to note that in a practical scenario, the entirety of the dataset would not be available to compute the most optimize threshold for the baseline method. We wanted to illustrate that accuracy is a much more reliable indicator even under the most optimal conditions for the baseline.
>
> CURL was not compared because it tackles a different learning setting where exposure boundaries are unavailable and not class-incremental. This comparison is unfair given that CURL addresses a harder learning environment and is therefore only able to achieve a performance of 77% on MNIST. The goal of our experiment section is to compare our method against unsupervised learning methods, not supervised. We chose the more up to date method, BiC, to serve as an oracle because it performs significantly better than iCarl.
>
> The ResNet-18 backbone was used for all baselines evaluated in the experimental section. We will update the manuscript to make this more clear.
>
> "After obtaining the correct label" is changed to "After obtaining the predicted label"
>
> Active learning models need to determine the lowest amount of supervision required to learn the task. In our case, we tackle a more difficult problem where no supervision is performed. Applying our method for active learning is a good suggestion for future work.

---

### Official Review · AnonReviewer4 · 2020-10-26
**A paper that I found difficult to read.**

**Rating:** 3
**Confidence:** 2

**Review:**

This article proposes a method for predicting whether a batch of data is of the same class as one of the classes already seen by a classifier or whether it contains data from another class. The idea is to then be able to incorporate this batch to the previous training set, in an unsupervised learning context. It is assumed that each batch contains data from only one class. Experiments are there to show the interest of this method for anomaly detection or incremental learning.

I find it difficult to formulate an opinion on this paper because I don't think I have managed to understand the detail of what is actually done. For example with regard to the detection of out of distribution data, the classic problem is whether a data is out of a distribution. Here it is not a data but a batch of data that is considered. I don't really see, under these conditions, how to compare to classical OOD methods.

As far as incremental classification is concerned, I don't understand the definition of the metric given in section 4.3 and therefore I'm not sure I understand what the task is really about. The fact that it's unsupervised makes it away from standard problems.

It seems to me that the paper lacks a clear definition of the tasks addressed and the means to evaluate performance.

---

> ### Author Response · Authors · 2020-11-12
> **Re: A paper that I found difficult to read**
>
> Thank you for your comments. Perhaps we can clear up some details.
>
> While classical OOD methods only address the novelty for single data points, these methods can be extended to determine novelty for a batch by taking the mean of the method-specific metric and comparing it to the OOD threshold. For example, the baseline method, IOLfCV, uses exposure average feature-distances to compare with a threshold. In the Appendix, we have mentioned that the mean of the method-specific metric was used "The mean MSP for all images belonging to an exposure is used to determine novelty". However, we will revise the manuscript to make this detail more explicit.
>
> During training, incoming exposures are presented incrementally. Each exposure contains data that belongs to a single class. Because iLAP does not have knowledge of the labels, pseudo-labels are assigned incrementally (0,1,2..). To evaluate the model's performance, a ground-truth mapping ($m$) is stored to map the pseudo-labels to the ground-truth class (0: cat, 1: dog, .....). However, because this is an unsupervised setting, it's possible for the model to falsely create multiple labels for a single class (0: cat, 1: dog, 2: cat ....). During inference time, if the model simply attributed each exposure to a new pseudo-label, accuracy would result to be 100% for the respective class. To prevent this, we divide the accuracy of the learner by the number of times the inverse mapping is repeated during inference ($\frac{1}{|m^{-1}(y)|}$). If the inverse mapping does not exist (the class was never detected by the model) the accuracy for the class is 0%. The total accuracy is the average of the class accuracies.

---

### Official Review · AnonReviewer3 · 2020-10-29
**Novelty detection via re-training**

**Rating:** 4
**Confidence:** 4

**Review:**

The paper studies an unsupervised class-incremental learning setting where a single class appears in each exposure, the classes can repeat and remain unknown during episodic training.  A set of exemplars is used to evaluate accuracy changes, based on which novelty is determined. The ideas is novel, but I is less scalable and the approach currently lacks key analysis and comparisons with the incremental learning methods and open-set approaches.

Pros`:
+ An novelty detection approach that considers the changes in accuracy of the previous tasks as a new task is learned by assigning a new label to the incoming episode. A threshold value is then used to detect novelty.

Cons:

- The basic intuition is that if a previous class is observed again, and the performance on old similar class will go down significantly. I feel this assumption is weak and can only be relevant in specialized cases, e.g., what if a very similar confusing class is observed? The currently evaluated datasets (SVHN, MNIST, CIFAR) do not consider such fine-grained cases.
- The evaluations in comparison to SOTA incremental learning methods is insufficient. Only a single approach, BiC, is considered for comparisons.
- The class imbalance based approach looks like a practical hack and is sensitive to the hyperparameters.
- The propose approach will incur a high computational cost with training the model at each episode to detect novelty. The computational cost comparison is not performed in the experimental section.
- The open set literature solves the same problem of OOD detection, however no comparison with SOTA methods is shown. I would recommend authors to check a nice survey on this topic: "Recent Advances in Open Set Recognition: A Survey"
- The paper is not well-written. Fig. 2 comes before class-imbalance discussion.

---

> ### Author Response · Authors · 2020-11-25
> **Re: Novelty detection via re-training**
>
> Thank you for your comments, you bring up valid concerns, and we hope that you find our comments sufficient.
>
> 1. Benchmarks are not fined-grained enough:
>
> While SVHN, MNIST have classes that are distinctly separable, CIFAR-100 contains several superclasses that each contain several similar classes. While an argument can be made that these individual classes are not fine-grained enough, our method was still able to significantly outperform the SOTA distance-based baseline with 96.5 out of the 100 classes discovered. In section 5.3 we describe why the re-training method would be more advantageous in the scenario with fine-grained classes. Re-training with a distinct label allows the learner to identify a separable feature space such that one exists.
>
> 2. SOTA incremental learning
>
> The goal of our experiment section is to compare our method against unsupervised methods, not supervised. The core contribution of our paper is to introduce a SOTA novelty detection method for an incremental setting and demonstrate its success in the unsupervised incremental learning environment. BiC serves as an oracle to demonstrate that our method was able to perform close to SOTA supervised methods despite the lack of labels.
>
> 3. Using class-imbalance as a hack
>
> We illustrate in the paper that there is merit in using class imbalance to maximize the performance drop between novel and repeated classes. We initialized the class-imbalance ratio based on 2 datasets (Fashion-MNIST, Imagenette) that were different from the benchmarks used for our evaluation (MINST, CIFAR-10, SVHN, CIFAR-100). These values were maintained  across all experiments and we have shown that it improves performance drastically.
>
> 4. Computation Cost
>
> Our method requires training each incoming exposure twice, first to detect novelty and then to perform normal IL after acquiring the predicted label. The $\mathcal{O}$ complexity remains the same because we are multiplying by a constant factor of two.
>
> 5. Open Set Recognition
>
> We have compared our method with various OOD detection methods. There are some methods in the OOD detection literature we excluded due to their unsuitability in the incremental learning environment. These methods require prior training on large datasets of closed classes which would not be suitable in our setting. [R1, R2]
>
> We have moved figure 2 to the suggested location.
>
>
> [R1] Neal, Lawrence, et al. "Open set learning with counterfactual images." Proceedings of the European Conference on Computer Vision (ECCV). 2018.
>
> [R2] I. Jo, J. Kim, H. Kang, Y.-D. Kim, and S. Choi, “Open set recognition by regularising classifier with fake data generated by generative adversarial networks,” IEEE International Conference on Acoustics, Speech and Signal Processing, pp. 2686–2690, 2018.

---

### Official Review · AnonReviewer1 · 2020-10-29
**interesting idea to tackle the unsupervised class-incremental learning but needs more experiments**

**Rating:** 6
**Confidence:** 4

**Review:**

This paper proposes to tackle the problem of unsupervised class-incremental learning, where the training data is composed of a sequence of "exposures". Each exposure is comprised of a set of images that pertains to a single class, where the class label is unknown while the boundaries between exposures are known. The key difficulty in such unsupervised class-incremental learning is to determine whether an arriving exposure belongs to what the classification model $L$ has learnt previously or is a novel one, thus relating to the problem of novelty detection. The proposed method address the novelty detection by an interesting idea: they always treat the current exposure as a novel class and use it to train the copy of classification model $\hat{L}$ together with the training exemplars of previously-learnt classes, if the current exposure actually belongs to one of the previous-learnt classes, the confusion occurs to make the classification accuracy significantly decrease (over a threshold) on that specific class, where the accuracy is computed based on the validation exemplars. Moreover, a technique of introducing class-imbalance into such confusion-based novelty detection is proposed and helps to boost the robustness of novelty detection.
There are some pros and cons of this paper as listed below.

Pros:
+ The idea of using confusion to address the novelty detection is novel and interesting, where the corresponding threshold is easier to be determined and contributes to better out-of-distribution performance in comparison to other related works of using static distance-based threshold.
+ The introduction of class-imbalance works well with the confusion-based novelty detection and its contribution is experimentally verified on various datasets.
+ The overall performance of the proposed method on unsupervised incremental learning is better than an unsupervised baseline (IOLfCV) and comparable to a supervised one (BiC).

Cons:
- The figure.2 is a little bit difficult for understanding the properties of seen and unseen classes with respect to class-imbalance ratio $\lambda$ at the first sight, e.g. why the curve of unseen classes would go up along with larger $\lambda$? Perhaps it is better to replace the terminology of "seen" and "unseen" classes by "repeated" and "non-repeated" classes?
- There is another closely-related type of incremental learning: unsupervised continual learning. Although its setting is more difficult than the unsupervised class-incremental learning which is tackled in this paper, it would still be nice to have the baselines of unsupervised continual learning for providing more insights to the readers.
- As the mostly-related work of this paper is Stojanov et al., CVPR 2019 (also addressing the unsupervised class-incremental learning problem), why the CRIB dataset proposed by Stojanov et al. is not used for evaluation here in order to have more direct comparison?
- Moreover, as indicated by Stojanov et al., the repetition of classes (e.g. how frequent a learnt class arrives again for learning) plays an important role for the model performance, there should be clear description on the experimental setting of repetition as well as the investigation on it in this paper.
- Furthermore, in the paper of Stojanov et al., they experiment with the classification models of having pre-trained feature extraction or being learnt from scratch. However, in this paper only the classification model pretrained on ImageNet is adopted. There should be experimental results and corresponding discussion on having the classification model trained from scratch for better understanding how the proposed confusion-based novelty detection behaves.
- Lastly, it is also important to investigate on the forgetting effect. When updating the classification with predicted label, are the techniques for avoiding catastrophic forgetting used (e.g. knowledge distillation)? If not, how the proposed method prevents the catastrophic forgetting from happening? If the forgetting does happen, will the confusion-based novelty detection still be working?

In brief, this paper proposes interesting idea of having confusion-based novelty detection approach to tackle the unsupervised class-incremental learning, but it needs more experiments and discussions to make the paper more complete and ready for ICLR. I would expect to see the concerns listed above being well addressed in the rebuttal.

---

> ### Author Response · Authors · 2020-11-25
> **Re: interesting idea to tackle the unsupervised class-incremental learning but needs more experiments**
>
> Thank you for your thorough and detailed review. We appreciate the feedback and will do our best to address your comments.
>
> Terminology for figure 2 has been changed to “repeated” and “non-repeated.”
>
> The class imbalance ratio represents the proportion of samples used per class from the exemplar versus the number of samples used from the incoming exposure. For example, a $\lambda$ of $.5$ could mean 60 images used from the incoming exposure, and 30 images per class from exemplars for detection training. At high values of lambda, the accuracies for non-repeated classes would decrease during detection training due to forgetting caused by insufficient samples of previous classes.
>
> We have evaluated our model using the CRIB dataset created by Stojanov et al. and have added the results to the experimental section of our manuscript. iLAP was able to perform similarly to IOLfCV in terms of net accuracy, but was able to incorporate ~7% more classes than the baseline. To further demonstrate the robustness of the class-imbalance ratio, we have utilized the same values, $\lambda = .5$ and $\theta = .6$.
>
> Repetition indeed plays an important role in model performance. Stojanov et al. found that given enough repetitions in the supervised case, the model will eventually reach batch performance. However, their method in the unsupervised setting was unable to reach batch performance. In section 5.2 we illustrate that the primary reason the baseline fails is not due to a higher number of repetitions, but rather the order in which the repetitions occur. In figure 3, as the model incorporates more and more classes, the distance between features decreases. Because the F-Score is computed over the entirety of the dataset, there is a high chance that the model will mistake repeated classes for non-repeated classes. This is a key reason for the substantial performance gains obtained from using our method. The shift in class-feature distance is largely overlooked by traditional distance-based OOD methods.
>
> We have included additional experiments where no pre-training is performed.
>
> Our method mitigates catastrophic forgetting by using replay. An exemplar for training is maintained at all times and samples are capped on a per-class basis (Section 3.1). Because a validation exemplar is also maintained, iLAP is able to self-evaluate its performance at any stage of training (using pseudo-labels). If the performance for a particular class is unable to be satisfied, the class is ultimately discarded to allow iLAP to relearn the class (Section 3.4).

---

### Decision · Program_Chairs · 2021-01-07
**Final Decision**

**Decision:**

Reject

**Comment:**

This paper presents a continual learning method based on a novelty detection technique. All reviewers are concerned about various issues, especially, motivation, experiment, and presentation. One of the reviewers was initially positive about this paper but downgraded his/her score due to unresolved problems in the proposed method. Considering all the comments and communications with the authors, AC believes that this paper is not ready for publication yet.